# Active Forms of Chemerin Are Elevated in Human and Mouse Ovarian Carcinoma

**DOI:** 10.3390/biomedicines13040991

**Published:** 2025-04-18

**Authors:** Lei Zhao, Qin Zhou, Venkatesh Krishnan, Justine Chan, Simone Sasse, Supreeti Tallapragada, Dan Eisenberg, Lawrence Leung, Oliver Dorigo, John Morser

**Affiliations:** 1Division of Hematology, Stanford University School of Medicine, Stanford, CA 94305, USA; lzhao02@stanford.edu (L.Z.); qzhou@stanford.edu (Q.Z.); lawrence.leung@stanford.edu (L.L.); 2Veterans Affairs Palo Alto Health Care System, Palo Alto, CA 94304, USA; de1@stanford.edu; 3Division of Gynecologic Oncology, Department of Obstetrics and Gynecology, Stanford Women’s Cancer Center, Stanford Cancer Institute, Stanford, CA 94305, USA; vkrishnan@stanford.edu (V.K.); jchan62r@stanford.edu (J.C.); ssasse@stanford.edu (S.S.); supreeti@stanford.edu (S.T.); odorigo@stanford.edu (O.D.); 4Department of Surgery, Stanford University School of Medicine, Stanford, CA 94305, USA

**Keywords:** chemerin, ovarian carcinoma, protease cleavage of chemerin

## Abstract

**Background:** Chemerin is a small adipokine that is activated and inactivated by proteolysis of its C-terminus with a role in regulating metabolism, immunity, and inflammation. Significant levels of chemerin are found in circulation and ascitic fluid of ovarian carcinoma patients. **Methods:** We investigated the levels of different chemerin forms in three cohorts: people with BMI < 25, with BMI > 40, and ovarian carcinoma ascites with ELISAs specific for different chemerin forms. Ascites from a mouse model of ovarian carcinoma were also analyzed, and the model was compared between wild-type and chemerin-deficient mice. **Results:** Conversion of plasma to serum increased the levels of processed chemerin with lower increases in samples from people with BMI < 25 than in people with BMI > 40. High levels of total chemerin and processed forms of chemerin were found in ascitic fluid from both patients who had a mean BMI of 29 and the mouse model. In chemerin-deficient mice the tumors grew slower than in wild-type mice. **Conclusions:** Serum has more processed and active chemerin than plasma irrespective of source. Ascites of ovarian carcinoma patients contained high levels of active chemerin, which, based on the mouse data, enhance tumor growth.

## 1. Introduction

In the developed world, epithelial ovarian cancer is the fourth most common cause of cancer-related death in women and has the highest mortality rate amongst all gynecological tumors [1]. In the USA it is estimated that there were ~20,000 new cases of ovarian carcinoma diagnosed in 2024, with ~13,000 women predicted to die from the disease [2]. The overall 5-year survival is ~50% depending on the type of ovarian cancer and the stage at which it is diagnosed. The reason for the high mortality rate is that many women are diagnosed at late stages and have inadequate treatment options, in particular, for platinum-resistant ovarian cancer [3]. Types of ovarian carcinoma are distinguished by the origin of the cancer, with epithelial adenocarcinoma being ~90% of cases. The most prevalent types consist of serous, endometrioid tumors and borderline and unspecified adenocarcinoma [4]. Ovarian cancer most commonly occurs in women after menopause [5].

Chemerin is a small protein that behaves as an adipokine and immune modulator [6,7]. It binds and signals via two receptors from the G-protein coupled receptor family, chem1 (chemokine-like receptor 1; CMKLR1) and chem2 (G protein-coupled receptor 1; GPR) as well as binding to a non-signaling receptor, CCRL2 before subsequent presentation to chem1 or chem2 [8,9,10,11].

In many tumors, chemerin is downregulated in the tumor tissue while elevated levels are observed in circulation [12]. In breast cancer, chemerin and chem1 are downregulated in the tissue, resulting in changes in the tumor micro-environment for persistent tumor growth while over-expression of chemerin reverses these effects [13]. In contrast, chemerin has been shown to promote the growth of several tumors, including clear-cell renal carcinoma and ovarian cancer [14,15,16]. Expression of chemerin correlated with progesterone receptor and estrogen receptor β when investigated by anti-chemerin antibodies staining of histological sections of ovarian carcinoma tissue. The level of chemerin determined by ELISA was higher in ascitic fluid than in serum in 12 ovarian carcinoma patients [17]. High levels of chemerin in ascitic fluid samples was also found to be high in two other studies [10,18].

Translation of chemerin mRNA results in a 163-amino-acid chain that has its signal peptide removed during secretion, resulting in a secreted form, chem163S, that is inactive [19]. Subsequent proteolytic processing of the C-terminus of the protein chain generates partially active chem158K, fully active chem157S, and chem156F before forming inactive chem155A and smaller forms. We specifically developed ELISAs for these varied forms of chemerin to detect both human and the homologous mouse versions to allow analysis of biopsy samples that always contain mixtures of these forms. In contrast, the commercial ELISAs used in most other publications do not distinguish between the different chemerin forms and have not shown that all chemerin forms have parallel dose response curves with equipotent signal for the different forms.

In this study, we hypothesized that ascitic fluid from ovarian carcinoma patients would contain high levels of active chemerin forms in comparison to the levels present in plasma from control individuals.

## 2. Materials and Methods

### 2.1. Human Ovarian Carcinoma Ascites Samples

The collection of ascitic fluid from ovarian carcinoma patients was approved by the Stanford Institutional Review Board under protocol numbers #42966. Ascites was directly collected from patients at the time of surgery or by paracentesis and transferred to the lab for post processing. After centrifugation of ascites samples at 300× *g* for 10 min, cell-free supernatant was immediately separated, aliquoted, and stored at −80 °C for further use. Upon thawing, protease inhibitors (Complete Protease Inhibitor, Roche Applied Science, Pleasanton, CA, USA) were added to the vials, while in a few samples, protease inhibitors were added immediately upon collection of ascitic fluid. The samples underwent one freeze-thaw cycle prior to chemerin measurements.

### 2.2. Human Plasma and Serum Samples

All studies were approved by the Stanford Institutional Review Board under protocol numbers #6946 and #24175, and all individuals were consented to participate in these studies. As controls, plasma and serum from volunteers were assayed for levels of the different chemerin forms. Factoring in the direct correlation of obesity to the high levels of chemerin available in circulation [20,21], we divided the donors into two cohorts based on a BMI < 25 vs. BMI > 40. Blood was collected from volunteers with BMI < 25 and from bariatric surgery patients (people with obesity; BMI > 40) into either serum or plasma tubes. The study volunteers were in good general health with age variation between 18 and 80 years old. The samples were acquired from the bariatric surgery patients immediately prior to surgery. Plasma and serum were prepared by centrifugation [19]. After centrifugation, both serum and plasma were immediately separated, aliquoted, and stored at −80 °C for further use. The samples underwent one freeze-thaw cycle prior to chemerin measurements. For the isolation of chemerin, plasma or serum was thawed, and 1 mL was mixed with 75 μL of heparin-agarose (Sigma, St Louis, MO, USA) and 43 μL of 25× Complete Protease Inhibitor (Roche Applied Science, Pleasanton, CA, USA). This mixture was shaken at 4 °C for 2 h before centrifugation to sediment the beads. After washing extensively with phosphate-buffered saline (PBS), the chemerin was eluted with PBS containing 0.8 M NaCl and the protease inhibitor cocktail.

### 2.3. Mouse Husbandry

Chemerin-deficient mice (chemerin KO) and wild-type mice (WT), both on C57Bl/6J background, were used in this study, as described previously [22]. Animal studies were performed at the Veterans Administration Palo Alto Health Care System (VAPAHCS), Palo Alto, CA, in accordance with National Institutes of Health guidelines. All procedures were approved by the institutional animal care and committee of VAPAHCS (protocol #LEU1566). Mice were housed together with ad libitum standard irradiated Teklad 2918 chow (Envigo, Indianapolis, IN, USA) and tap water under a 12-h light/dark schedule.

### 2.4. Mouse Model of Ovarian Carcinoma

Murine ovarian cancer cells (ID8), derived from ovarian epithelium from C57BL/6J mice [23], were cultured at 37 °C with 5% CO_2_ in DMEM with 100 U/mL penicillin in 5% fetal bovine serum supplemented with insulin, transferrin, and selenium (25-800-CR, Corning, Corning NY, USA) [24]. Cells were collected by trypsinization, centrifuged, and resuspended in PBS. 5 × 10^4^ cells were administered i.p. into 6-week-old female mice. Some experiments included groups of mice injected with PBS (chemerin KO con and WT con). Mice were weighed twice weekly until accelerated wight gain was observed, after which they were weighed daily. Mice were sacrificed when the weight gain reached the limit specified in the IACUC protocol. Ascitic fluid was collected by aspiration, centrifuged, and the supernatant frozen. The comparison of growth of Id8 tumors in chemerin KO mice to WT mice was repeated three separate times.

### 2.5. ELISAs

The panel of ELISAs for different chemerin forms for both human and mouse has been designed as previously described and includes an ELISA that detects all forms but with varying potency and specific ELISAs for five individual forms [19]. The nomenclature for human chemerin forms (mouse forms in brackets) is the C-terminal amino acid number followed by its one letter code: the precursor is referred to as chem163S (mProchem), the partially active form as chem158K (mchem157R), the active forms as chem157S (mchem156S) and chem156F (mchem155F), and the inactive form as chem155A (mchem154A) [19]. Cleaved chemerin forms were calculated by subtracting the level of the precursor form from total chemerin forms. Degraded chemerin forms were calculated by subtracting the sum of the specific chemerin forms (in human: chem163S, chem158K, chem [157S + 156F], and chem155A; in mouse: mProchem, chem157R, mchem [156S + 155F], and mchem154A) from total chemerin forms.

### 2.6. Statistics

No outliers, defined as having a value > 3SDs away from the mean, were present in the data. Area under curve (AUC) for % weight gain in the ID8 mouse model was calculated using the trapezoid rule with the baseline set to 100%. Data on the levels of the different chemerin forms and AUC of the % weight gain were analyzed by ANOVA followed by post hoc Tukey–Kramer test in Prism GraphPad v10.2.2 running on a MacBook Air (OS v14.6.1). Comparison of survival curves was performed by Kaplan–Meier analysis with log rank test for four groups, and the data on chemerin KO was compared to WT by unpaired student *t* test. Results are presented as either ± standard error of the mean (SEM) or ± standard deviation (SD). Statistical significance was determined based on (*p* < 0.05) and is represented in figures.

## 3. Results

### 3.1. Levels of Chemerin Forms Change When Plasma Is Converted to Serum

In order to provide baseline levels of the different chemerin forms for comparison with those found in ascitic fluid from ovarian carcinoma patients, we collected plasma from participants with BMI < 25, n = 22 and patients with obesity undergoing bariatric surgery (>40 BMI; n = 22). The demographics of the participants are shown in Table 1, and their medications are shown in Appendix A. The two groups were matched in age and gender while the race/ethnicity differed between these groups. People with obesity were taking more medications than the BMI < 25 volunteers.

Serum was derived from plasma samples immediately after blood collection before freezing. Total chemerin was measured by an ELISA that detects all chemerin forms, albeit with different sensitivities [20]. The levels of five different chemerin forms (chem163S, chem158K, chem157S, chem156F, and chem155A) were measured by specific ELISAs, and the amounts of cleaved and degraded chemerin were calculated [25].

In all samples, changes in the levels of all the different chemerin forms were observed when plasma was clotted to generate serum (Figure 1). In general, the levels of chemerin changed in the same direction in all the samples from both cohorts.

In the plasma and serum samples from volunteers with <25 BMI, there was more total chemerin, chem155A, cleaved chemerin, and degraded chemerin in serum than in plasma and less chem158K (Figure 2a, Table 2). There was no change in the level of chem163S and chem157S; however, there was a trend to an increase in chem156F. In both groups, the levels of cleaved and degraded chemerin increased dramatically.

In individuals with a BMI < 25, the levels of total chemerin, cleaved chemerin, degraded chemerin, chem157S, chem156F, and chem155A were lower compared to people with BMI > 40, while levels of chem163S and chem158K were higher (Figure 2, Table 2 and Table 3).

Samples from people with obesity contained more total chemerin than the BMI < 25 group, irrespective of whether plasma or serum were assayed (Table 4) [21]. Chem163S was marginally lower in plasma from the BMI > 40 group while similar in serum from both groups. In addition, chem158K was unchanged in people with obesity. All the other chemerin forms were increased in people with obesity except for chem158K in serum. Levels of chem157S and chem156F, the active chemerin forms were both higher in plasma in people with obesity than from BMI < 25 volunteers, and chem157S was also increased in serum.

### 3.2. Levels of Chemerin Forms in Human Ovarian Carcinoma Ascites

Samples of ascitic fluid were collected from patients diagnosed with ovarian carcinoma. The demographic characteristics of the ovarian carcinoma patients are presented in Table 5, and their medications and treatments are shown in Appendix A. Accumulation of ascites is commonly associated with different types of ovarian carcinoma and is often used as an indicator for initial diagnosis and tumor relapse [26,27,28].

In order to investigate if the levels of different chemerin forms were being modified by ex vivo proteolytic processing, we determined the different chemerin forms in ascitic fluid from ovarian carcinoma patients in the absence or presence of protease inhibitors (Figure 3). In the samples analyzed, there was minimal change in the samples processed without immediate addition of protease inhibitors upon sample acquisition. Based on this observation, we were able to evaluate archived samples that had been stored without protease inhibitors, with protease inhibitors added upon thawing.

Total chemerin was measured by ELISA for several different chemerin forms on 19 samples (OC) of ascites collected from women with ovarian carcinoma. Individual levels of total chemerin varied from undetectable in one sample to >90 ng/mL (Figure 4a). The mean level (± SD) was 48.3 ± 26.1 ng/mL, similar to the level of total chemerin in plasma [20,29]. There were no apparent differences between ovarian carcinoma with different sites of origin (fallopian tubes, ovaries, and peritoneum), for which several samples were available. The levels of total chemerin in ascites from other tumors were similar to those from ovarian carcinoma. The variability was unaltered when the chemerin was normalized to the protein content of the ascites (mean level ± SD: 1.23 ± 0.57 ng/mg) but the rank order was changed, suggesting that the exudation of fluid was occurring at different rates in different patients (Figure 4b). One sample was obtained from a patient with a benign cyst in which the total chemerin level was lower but was similar to the malignant samples when compared on a chemerin/protein mass.

Five specific ELISAs were run on these samples to measure the levels of different chemerin forms generated during chemerin activation and inactivation. High levels of one of the active forms of chemerin, chem157S (mean level ± SD: 19.0 ± 12.3 ng/m), in all of the samples while high levels of the other active form of chemerin, chem156F (mean level ± SD: 14.3 ± 21.4 ng/m), were present in some samples when the individual forms were analyzed by specific ELISAs (Figure 5). The levels of chem157S and chem156F were high enough to interact significantly with the chemerin receptors, chem1 and chem2 [30,31]. In some samples chem157S and in others chem156F predominated. The average BMI of the ovarian carcinoma patients at the time the samples of ascites were acquired was 29 ± 1.7, placing this group as overweight. The distribution of the different chemerin forms in these ovarian carcinoma ascitic samples differ from those in plasma or serum, whether from participants with BMI < 25 or patients with BMI > 40 (Figure 1) in the fraction of active chemerins and the low proportion of intact precursor chem163S.

Cleaved chemerin represents the level of the sum of all forms of chemerin smaller than the precursor, chem163S. When cleaved chemerin, calculated by subtracting the value of chemerin 163S form from total chemerin forms, the level of processing of the precursor, chem163S, varied from relatively little to complete proteolysis with mean levels of cleaved chemerin ± SD: 40.5 ± 28.6 ng/mL (Figure 6a). This is much higher than that found in normal plasma or serum, suggesting there is more proteolytic activity in the ascitic fluid of ovarian carcinoma patients than even in serum (Figure 2).

Degraded chemerin forms represent the chemerin forms smaller than chem155A that are inactive on chem1 and chem2. Degraded chemerin forms were calculated by subtracting the sum of the specific chemerin forms (chem163S, chem158K, chem [157S + 156F], and chem155A) from total chemerin forms (Figure 6b). The calculated levels of degraded forms varied from undetectable to levels greater than that determined by the total ELISA (mean level ± SD: 31.6 ± 34.4 ng/mL). This apparently surprising result can occur because the sensitivity of the total ELISA to some of the degraded forms may be much lower than to other forms [20].

The mean levels of different forms in ascitic fluid from ovarian carcinoma patients were calculated with higher levels of chem157S and chem156F, the active forms of chemerin, than in plasma (Figure 7a and Table 6). All samples had more proteolytically cleaved forms of chemerin than are in plasma or serum (Figure 2 and Figure 3). All samples contained high enough levels of the two fully active chemerin forms (chem157S and chem156F) for them to interact with the chemerin signaling receptors, chem1 and chem2. Chem157S predominated in some samples that also had undetectable chem156F while in others chem156F was the major form, but in those cases chem157S was also present.

The % distribution of the different chemerin forms in ovarian carcinoma ascites were calculated (Figure 7b). In most samples, the majority of chemerin molecules had been processed into active chem157S and chem156F, the active forms of chemerin, in contrast to plasma and serum (Figure 2). The inactive precursor, chem163S, is the overwhelming major form in plasma and serum, while in ascites from ovarian carcinoma patients chem163 is a minor form with the exception of only two samples, OC153 and OC200.

### 3.3. Levels of Chemerin Forms in Mouse Ascites

ID8 is an ovarian carcinoma cell line that, when inoculated peritoneally, develops ascites in a syngeneic mouse model. Ascitic fluid was harvested from a cohort of mice upon sacrifice before assaying the levels of total chemerin as well as a panel of ELISAs that measure specific forms of chemerin [32]. Total chemerin and the different forms were measured by ELISA in these samples (Figure 8 and Table 7). Total chemerin was present at (mean level ± SD) 110 ± 24.0 ng/mL There were low levels of the precursor (mean level ± SD: 18.2 ± 4.62 ng/mL) but high levels of mchem157R (mean level ± SD: 73.9 ± 8.43 ng/mL), which is homologous to the partially active human chem158K. For the two active forms, mchem156S and mchem155F that are homologous to human chem157S and chem156F, we observed low levels of mchem156S (mean level ± SD: 4.51 ± 2.05 ng/mL) but higher levels of mchem155F (mean level ± SD: 28.4 ± 12.1 ng/mL). The inactive mchem154A (homologous to human chem155A) was detected at a low level (mean level ± SD: 6.84 ± 2.07 ng/mL). Cleaved chemerin (total chemerin–precursor) was at a high level (mean level ± SD: 92.1 ± 20.5 ng/mL), but the calculation of the amount of degraded chemerin (total chemerin–[mProchem + mchem157R + mchem156S + mchem155F + mchem154A]) indicated no detectable levels. Taken together, more than 80% of mouse chemerin had been proteolytically processed in the ascitic samples.

The major form of chemerin in ID8 tumor-induced ascites in mice is the partially active precursor, mchem157R (Figure 7b). The form with the next highest level is active mchem155F, which is present at a concentration sufficient to trigger signaling on chem1 and chem2 followed by the unprocessed precursor, mProchem, while the other active form, mchem156S, and inactive mchem154A are present at low levels.

### 3.4. Development of Ascites in WT and Chem KO

Based on the above results demonstrating that chemerin is highly proteolytically processed in ascites from both human patient samples and in the ID8 mouse tumor model, we investigated if the lack of chemerin would affect the outcomes in the ID8 mouse model. Chemerin KO mice and WT mice were followed by weight until sacrifice. The time of sacrifice depended on the maximum allowable weight gain as determined by the IACUC or at a maximum of 14 weeks after inoculation (Figure 9a). Chemerin KO con and WT con mice were included to check that the weight gain observed for chemerin KO and WT mice with ID8 tumors was greater than the expected weight gain for mice over 14 weeks. None of the control mice died during the course of the experiment. Chemerin KO (n = 21) had a longer median survival time of 79 days compared to 72 days for WT mice (n = 24; *p* = 0.0449) with a hazard ratio of 2.12 (95% confidence interval: 1.02–4.37) (Figure 9b).

WT mice gained more weight than chemerin KO mice, and both gained more weight than either chemerin KO con or WT con mice (Figure 9a). The data was quantified by comparing AUCs and showed that WT AUC (mean ± SEM: 983 ± 101 weight gain %.days) was different from chemerin KO (mean ± SEM: 578 ± 99 weight gain %.days; *p* = 0.0188), WT con (mean ± SEM: 371 ± 73 weight gain %.days, *p* = 0.0020) and chemerin KO con (mean ± SEM: 558 ± 54 weight gain %.days; *p* = 0.0008) but that ID8 growth only caused a trend to an increase in weight gain in chemerin KO compared to the chemerin KO (Figure 9c). After the sacrifice of the first mice in this study cohort at Day 66, weight measurements were distorted based on survival bias. Chemerin-deficient mice have reduced ovarian carcinoma in this model, suggesting that chemerin is enhancing tumor growth.

## 4. Discussion

Chemerin is secreted from cells as an inactive precursor (chem163S) that requires proteolytic cleavage into functional forms that can further activate its specific receptors, chem1 and chem2 [33]. In addition to the two active forms, chem157S and chem156F, several other forms have been identified in biopsy samples, including partially active chem158K and inactive chem155A and chem144D [34]. Here, we show that conversion of plasma to serum in individuals with <25 BMI results in changes in total chemerin as well as many of the different chemerin forms (Table 2). This is explained by the activation of enzymes in the coagulation cascade that can also process chemerin as well as by release of chemerin from platelets [33,35,36]. Because we did not measure the chemerin content of the platelets, we are unable to quantitate how much chemerin or which chemerin forms contributed to the levels in serum.

In this study, we identified a higher fraction of cleaved chemerin in plasma than in our earlier reports [20,30]. Since these prior studies, we have developed ELISAs for other forms, in particular, chem156F, which account for the increase we observed in proteolytic processing.

In the bariatric surgery patients with >40 BMI the changes upon converting plasma to serum were larger than in individuals with BMI < 25 (Table 3 and Table 4). This is consistent with the hypothesis that inflammation is a part of the mechanism by which obesity develops and, thus, a low level of activation of inflammatory proteases is present in the circulation of individuals with high BMI, leading to increased proteolytic cleavage of chem163S [37].

The differences in levels of the chemerin forms detected in circulation are easiest to explain as a consequence of differences in metabolism between people with BMI < 25 and those with BMI > 40 as well as other differences between the groups that might confound the results. The race/ethnicity in the BMI < 25 cohort included a higher fraction of Asians and fewer whites than the BMI > 40 cohort that might affect the comparison.

Both race and age are potential confounding factors to the results here. And the size of the groups is too small for a definitive analysis of race and age. To our knowledge, there have been no studies of the effects of race on chemerin levels, nor have there been any longitudinal studies on chemerin levels. When a group of healthy centenarians was compared to a group that was 70–80-year-olds, the centenarians had a lower total chemerin level than the younger group [38]. In contrast, chemerin was found to increase with age in a study on the use of chemerin levels as a predictor of acute coronary syndromes [39]. In a comparison of chemerin levels in serum to those in cerebrospinal fluid (CSF) in patients with neurological disorders, no correlation with age was found in serum but was in CSF [40]. The mean age was similar in the two cohorts of people with BMI > 40 (47 ± 1.6) and BMI < 25 (47 ± 0.4), suggesting that chemerin differences due to age was not affecting the comparison between these two groups but was significantly older in the ovarian cancer patient cohort (63 ± 1.6).

The fractions of the different chemerin forms found in ovarian carcinoma ascitic samples with a low proportion of intact precursor chem163S plus high levels of active chemerin, chem157S and chem156F, differ from those in circulation irrespective of whether they were plasma or serum and if they were from participants with BMI < 25 or patients with BMI > 40 (Figure 1) in the low proportion of intact precursor chem163S. They are also different from the levels in synovial fluid and cerebrospinal fluid where chem158K, the inactive partially processed form, is the major form and in plasma, where the intact precursor predominates [29].

Analysis of the levels of different chemerin forms in ascitic fluid from ovarian carcinoma patients showed high levels of proteolytic processing of chemerin in all samples with an average of >80% cleavage (Figure 7 and Table 6). This is much higher than in the plasma and serum samples, where the maximum cleavage is <40% found in plasma samples from people with obesity (Figure 2c). By comparing samples of ascitic fluid collected in the presence or absence of protease inhibitors with a broad spectrum of inhibition, we showed no differences in levels of chemerin forms showing that the chemerin forms had not been generated by ex vivo processing (Figure 3). The level of chemerin proteolytic cleavage found suggests that the ovarian carcinoma is a highly inflammatory environment [41,42,43]. Previously, we have found such high levels of chemerin cleavage in biopsy samples from other indications with inflammation, such as synovial fluid from arthritis patients or cerebrospinal fluid from glioblastoma patients [25,29]. In addition, various classes of proteases including kallikreins, coagulation, and fibrinolytic enzymes that can process chemerin have been identified in ascites [44,45,46,47].

The active chemerin forms, chem157S and chem156F, have ED_50_s around 5 nM on chem1 and chem 2 [30,31]. In ovarian carcinoma ascitic fluid chem157S and chem156F are present at levels that would cause activation of the chemerin receptors and downstream chemerin effects and is consistent with chemerin modulating the disease. The volume of ascites fluid in ovarian carcinoma can reach >2 L, which would contain ~100 mg of total chemerin [48].

Most patients in this study were diagnosed with high-grade serous ovarian carcinoma, but no differences were found with respect to their sites of origin. A few samples from different types of ovarian carcinoma were included in this study as well as ascites from patients with other cancers, but no differences were observed in the levels of chemerin forms. This is a very tentative conclusion because the numbers of patients were low.

Id8 cells are a well-accepted model of ovarian carcinoma that has been widely used [49,50]. When injected intra peritoneally into mice, ID8 cells form ascites similar to human ovarian carcinoma ascites [23,51]. We measured the levels of different chemerin forms in ascitic fluid from this murine model of ovarian carcinoma, finding that there were high levels of proteolytic cleavage of chemerin, including the active forms, mchem156S and mchem155F (Figure 8, Table 7). The levels of the active forms were sufficient to trigger responses at the mouse chemerin receptors. The distribution of the different chemerin forms was similar to that found in ascites from human ovarian carcinoma patients, suggesting that this model would be suitable for testing the role of chemerin.

Unsurprisingly, the coefficient of variance (CV) of cleaved chemerin and degraded chemerin was high because they were calculated from the results of multiple ELISAs. Overall CVs were lowest in the data sets of chemerin levels in the ascites fluid from the mouse ID8 model, which were calculated to provide a comparator for the human data. The CVs in chemerin ELISAs determined on samples from the Id8 model would be expected to be more consistent as the genetic background of the mice was identical, the mice were housed and fed identically, and ID8 cells are a cloned cell line with very little variation. In general, the CV was higher in serum samples than in plasma samples, suggesting that, when possible, chemerin and its forms should be analyzed in plasma. The CV was similar in samples from both <25 and >40 BMI cohorts of participants irrespective of their BMI. The CV was highest in the ascitic fluid samples from ovarian cancer patients possibly as this a more heterogeneous cohort than the other two cohorts with different types of ovarian cancer at various stages of the disease with differing treatments.

Based on the data on the high levels of proteolytic cleavage of chemerin in ascites from the ID8 model, we investigated the role of chemerin in ID8 tumor growth by comparing chemerin deficient to WT mice. ID8 tumor growth was slower in chemerin KO mice than in WT mice, leading to their prolonged survival (Figure 9). Therefore, we confirmed that the production and activation of chemerin drives tumorigenesis in the ID8 model which is consistent with reports in some other models but is in contrast to others [12]. This conclusion that murine ovarian carcinoma growth is enhanced by chemerin is consistent with the surprisingly high levels of active chemerin found in ovarian carcinoma ascites, suggesting that chemerin is also exacerbating tumor growth. Although the Id8 mouse tumor model represents well one type of ovarian cancer, it does not represent the various other types and also, has all the well-documented limitations of mouse models for translation to humans. In particular, the Id8 model uses a mouse cell line that does not represent the genetic heterogeneity found in all human tumors while the mouse immune system has differences from the human immune system [52].

In addition to chemerin’s regulation of the immune system regulating the host anti-tumor response [6,7,18,53], chemerin also regulates tumor growth via its effects on energy balance and glucose and lipid metabolism [54,55,56]. The tumor micro-environment contains immune cells, endothelial cells, fibroblasts, and adipocytes amongst cells mobilized by the tumor cells. Ovarian carcinoma reprograms adipocytes to mobilize their energy stores by changing their energy balance [57]. These changes in energy metabolism, which chemerin could be regulating, contribute to the Warburg effect, possibly leading to resistance and more aggressive growth [58,59,60].

Chemerin can both enhance or inhibit cancer growth, which has been reviewed for ovarian carcinoma amongst other cancers [12,61]. In a mouse model of clear cell renal carcinoma, for example, inhibition of chemerin by an anti-chemerin antibody improves outcomes via both chem1 and chem2 that, in turn, regulates lipid metabolism [14,62]. Pharmacological or genetic suppression of either chem1 or chem2 improved outcomes although the mechanistic pathways differed. In patients with oral squamous cell carcinoma, higher serum levels of chemerin correlate with worse outcomes [63]. In squamous cell carcinoma cell lines, chemerin activates STAT-3 signaling, increasing production of IL-6 and TNF-α, leading to infiltration of neutrophils promoting tumor growth [64].

Chemerin can also inhibit tumor growth, as demonstrated in studies including melanoma and hepatocellular carcinoma. In the B16 cell mouse model of melanoma, chemerin expression reduced tumor growth in vivo but not in vitro, showing that the effect required the host immune system or other cells present in the tumor micro-environment [65]. Similarly, in hepatocellular carcinoma chemerin had a protective role by reducing the number of myeloid-derived suppressor cells [65,66]. Overall, the contradictory effects of studies on the role of chemerin in different cancers may be because the composition of the tumor micro-environment is different in different cancers. The importance of adipocyte reprogramming and changes in energy metabolism as well as in the host anti-tumor immune response varies among cancers, suggesting that chemerin’s modulation of these elements may result in the apparently contradictory data.

Obesity is recognized as a risk factor for cancer with adipose tissue providing a source of secreted factors referred to as adipokines in addition to the role of adipose tissue in metabolism [67]. As well as regulating energy balance and other metabolic parameters, adipokines modulate the immune system and inflammation and affect ovarian carcinoma [61,68]. In addition to its role as an adipokine, chemerin, like other adipokines, regulates the immune system and uniquely as an adipokine can increase blood pressure [69,70].

In cancer, chemerin has both tumor-promoting via direct effects on the cancer cells and the tumor microenvironments and tumor-suppressing roles mostly via the immune system [18]. Chemerin treatment enhances migration and invasion of ovarian carcinoma cell lines by activation of chem1 leading to epithelial to mesenchymal transition (EMT) in cell culture experiments [71]. In two ovarian carcinoma cell lines, one with high metastatic potential and the other as the parental cell line, the parental cell line produces less chemerin, suggesting that higher levels of chemerin enhances metastasis [17]. In these two cell lines, treatment with chemerin promotes proliferation and induces expression of PD-L1 inhibiting host anti-cancer immune reactions. In different ovarian carcinoma cell lines, treatment with chemerin inhibited proliferation-induced bisphenol-A and its derivatives [72].

Ovarian carcinoma uses active chemerin to enhance growth, suggesting that chemerin antagonism is a therapeutic target. In particular, the chemerin receptors, chem 1 and chem2, as GPCRs are in a class of proteins for which many other drugs have been developed [73,74]. One small molecule antagonist reported to date of chem1 is 2-(alpha-naphthoyl) ethyltrimethylammonium iodide (alpha-NETA), which does not have good specificity [75]. Other small molecules and peptide analogs that are antagonists for chem1 or chem2 have been described [76,77,78]. In addition, other approaches to antagonizing chemerin signaling have been evaluated, including antibodies directed against chem1 and anti-sense oligonucleotides to inhibit chemerin production [79,80]. Although these approaches have shown promising data in animal models, so far none have advanced into the clinic, and no trials are listed in clinicaltrials.gov.

Ovarian carcinoma cells increase interferon alpha (IFN-α) production while in patients overall survival correlates with increases in IFN-α and interferon-induced genes [81]. In cell lines chem1 but not chem 2 is well-expressed on all ovarian carcinoma cell lines investigated, suggesting that active chemerin binds to chem1 to signal into the cells. In tumor samples, immunohistochemistry showed the presence of both chemerin and chem1 and an association with steroid hormone receptors [16]. Chemerin reduces IGF1 production by human granulosa cells as well as activating the MAPK pathway and AKT [82]. Additionally, patients with high chemerin mRNA had lower overall survival than those with low chemerin mRNA. Studies of the chemerin signaling mechanism via chem1 in ovarian carcinoma cells are required to improve understanding of the reported chemerin effects. Future studies also should focus on the role of chemerin in regulating different immune cells in ovarian carcinoma as well as the effects of chemerin on the cells that form the tumor micro-environment, including fibroblasts, adipocytes, and endothelial cells.

## 5. Conclusions

We showed that conversion of plasma into serum changes the levels of different chemerin forms because of proteolytic processing, which underlines the importance of sample handling in future studies. There were higher levels of the proteolytically cleaved chemerin forms in ovarian carcinoma ascites than in samples either of the other two cohorts of participants with normal BMI (<25) or people with obesity with BMI > 40, showing that the ascitic environment is inflammatory, allowing generations of physiological-relevant levels of active chemerin. In the murine context, the mouse ID8 model recapitulates aspects of ovarian carcinoma, including the inflammatory milieu of the ascitic fluid, causing cleavage of chemerin. Taken together, in the murine ID8 model, we show that chemerin acts as pro-tumorigenic element. Overall, this study shows that chemerin enhances ovarian tumor growth in both human and mouse.

## Figures and Tables

**Figure 1 biomedicines-13-00991-f001:**
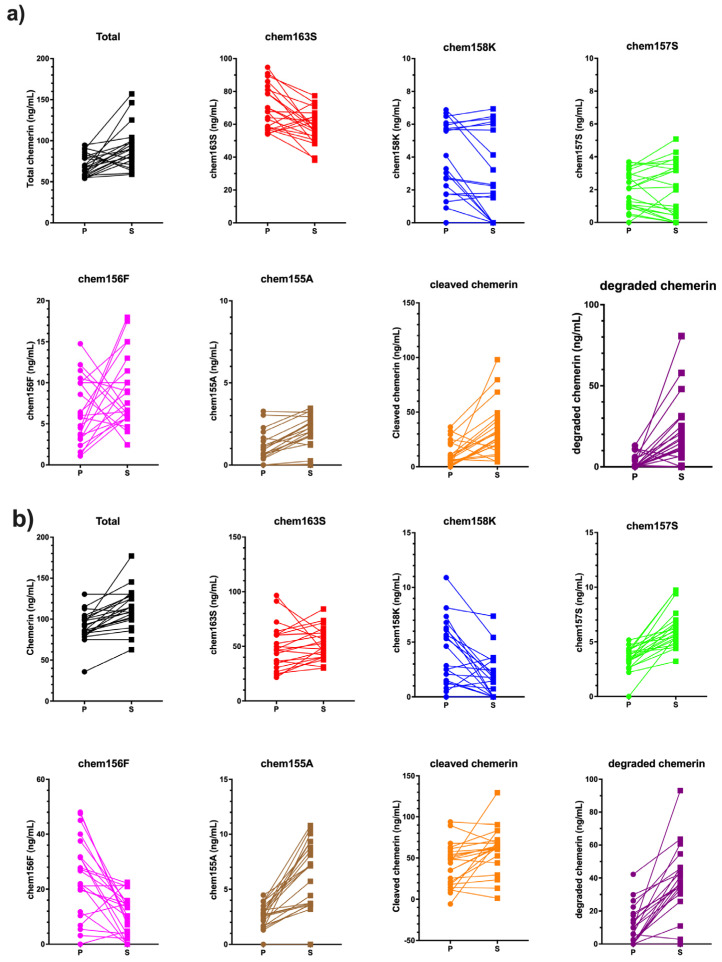
Changes in levels of chemerin forms in plasma and serum from patients with BMI < 25 and patients with BMI > 40. Blood was obtained, and either plasma or serum was used prior to measurement of chemerin by ELISA. (**a**) levels of different chemerin forms in plasma and serum from volunteers < 25 BMI (N = 22) (**b**) levels of different chemerin forms in plasma and serum from people with obesity with BMI > 40 (N = 22). P: plasma, S: serum.

**Figure 2 biomedicines-13-00991-f002:**
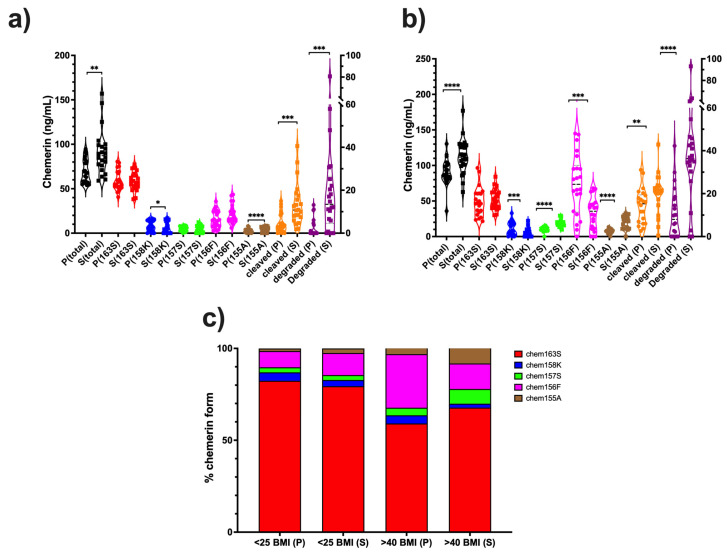
Changes in the levels of chemerin forms in the plasma and serum from two cohorts, BMI < 25 and BMI > 40 (each group N = 22), as determined by ELISA. Blood was obtained and either plasma or serum was prepared before measuring levels of chemerin by ELISA. P: plasma, S: serum. (**a**) levels of different chemerin forms in plasma and serum from volunteers with BMI < 25. Total chemerin, chem163S, and cleaved chemerin plotted on the left Y axis, chem158K, checm157S, chem156F, chem155A, and degraded chemerin plotted on the right Y axis. (**b**) levels of different chemerin forms in plasma and serum from people with obesity (BMI > 40). Total chemerin, chem163S, and cleaved chemerin plotted on left Y axis, chem158K, checm157S, chem156F, chem155A, and degraded chemerin plotted on right Y axis. (**c**) Average % of different chemerin forms in plasma and serum from volunteers and people with obesity. Data was analyzed by two-tailed paired Student *t* test between plasma and serum samples. * *p* < 0.05, ** *p* < 0.01, *** *p* < 0.001, **** *p* < 0.0001.

**Figure 3 biomedicines-13-00991-f003:**
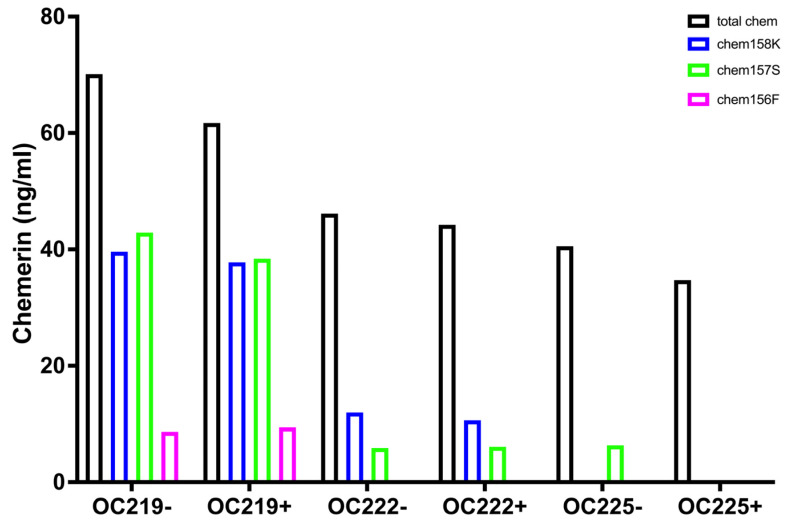
Inclusion of protease inhibitors does not significantly affect levels of chemerin forms in ascitic fluid samples. Samples from ascitic fluid of three ovarian carcinoma patients were processed with (+) or without (−) the addition of protease inhibitors before measuring levels of chemerin by ELISA.

**Figure 4 biomedicines-13-00991-f004:**
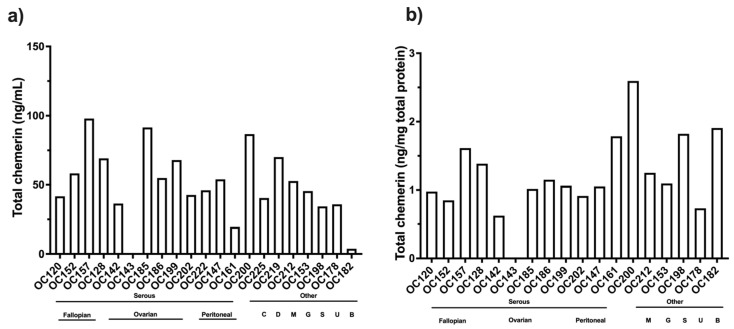
Levels of total chemerin (total chem) in ascites from patients with ovarian carcinoma determined by ELISA (N = 19) (**a**) total chemerin/mL (**b**) total chemerin normalized to protein content of ascitic fluid. Samples marked as Serous and then by site of origin, Fallopian (Fallopian tube serous carcinoma), Ovarian (Serous ovarian carcinoma), Peritoneal (Primary peritoneal serous carcinoma) and other marked as C (clear cell ovarian carcinoma), D (de-differentiated ovarian carcinoma), and M (mucinous borderline ovarian carcinoma). Also included are four ascites samples from three patients with other cancers, G (gastric signet adenocarcinoma), S (cervical squamous adenocarcinoma), and U (uterine carcinosarcoma), and one benign, B (benign serous cyst, para-ovarian).

**Figure 5 biomedicines-13-00991-f005:**
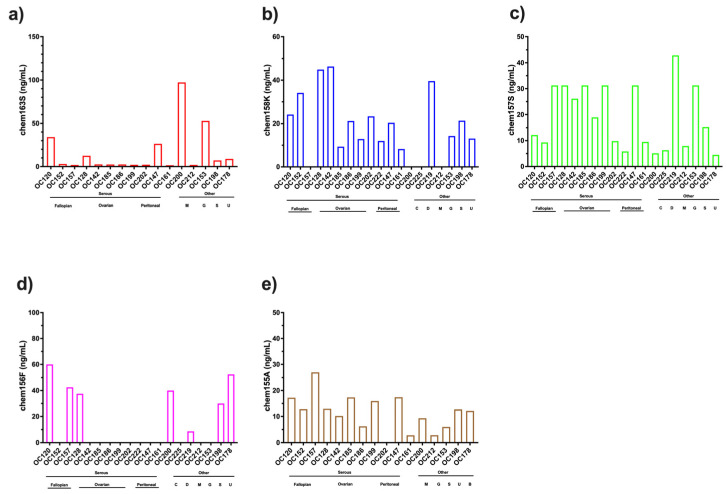
Levels of different chemerin forms in ascites from patients with ovarian carcinoma (**a**) chem163S (N = 16) (**b**) chem158K (N = 19) (**c**) chem157S (N = 19) (**d**) chem156F (N = 19) (**e**) chem155A (N = 16). Samples marked as Serous and then by site of origin, Fallopian (Fallopian tube serous carcinoma), Ovarian (Serous ovarian carcinoma), Peritoneal (Primary peritoneal serous carcinoma) and other marked as C (clear cell ovarian carcinoma), D (de-differentiated ovarian carcinoma), and M (mucinous borderline ovarian carcinoma). Also included are three ascites samples from patients with other cancers, G (gastric signet adenocarcinoma), S (cervical squamous adenocarcinoma), and U (uterine carcinosarcoma).

**Figure 6 biomedicines-13-00991-f006:**
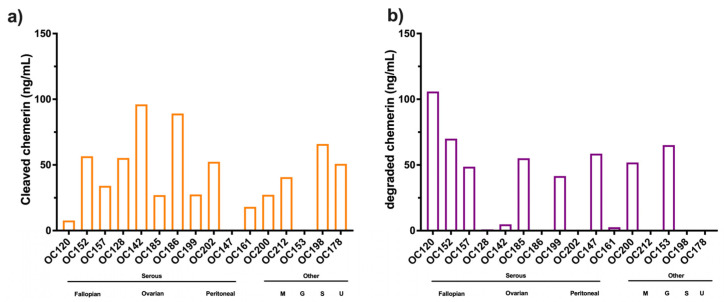
Levels of cleaved and degraded chemerin forms in ascites from patients with ovarian carcinoma (N = 16) (**a**) cleaved chemerin calculated as total chemerin–chem163S (**b**) degraded chemerin calculated as total chemerin–(chem163S + chem158K + chem157S + chem156F + chem155A). Samples marked as Serous and then by site of origin, Fallopian (Fallopian tube serous carcinoma), Ovarian (Serous ovarian carcinoma), Peritoneal (Primary peritoneal serous carcinoma), and one other marked as M (mucinous borderline ovarian carcinoma). Also included are three ascites samples from patients with other cancers, G (gastric signet adenocarcinoma), S (cervical squamous adenocarcinoma), and U (uterine carcinosarcoma).

**Figure 7 biomedicines-13-00991-f007:**
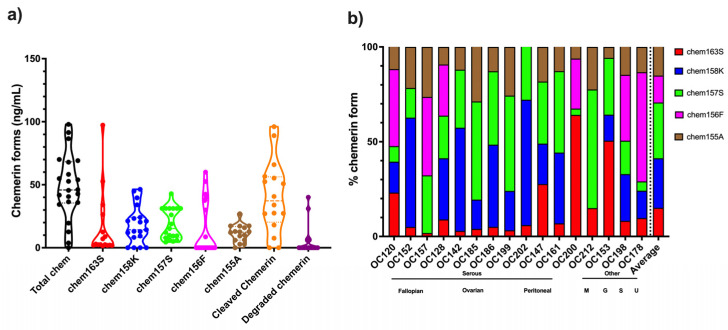
Levels of different chemerin forms in ascites from patients with ovarian carcinoma determined by ELISA for total chemerin and specific forms. (**a**) Values of different forms for overall cohort. Data shown in a violin plot with thick dotted line representing the median and the thin dotted lines quartiles. N = 16 except for total chemerin where N = 22 and chem158K, chem157S, and chem156F where N = 19. (**b**) The % distribution for each specific chemerin form in patients for which complete ELISA data is available (N = 16) is displayed based on data in Figure 4a and Figure 5. The average % distribution is shown in the right-most column. Samples marked as Serous and then by site of origin, Fallopian (Fallopian tube serous carcinoma), Ovarian (Serous ovarian carcinoma), Peritoneal (Primary peritoneal serous carcinoma), and one other marked as M (mucinous borderline ovarian carcinoma). Also included are three ascites samples from patients with other cancers, G (gastric signet adenocarcinoma), S (cervical squamous adenocarcinoma), and U (uterine carcinosarcoma).

**Figure 8 biomedicines-13-00991-f008:**
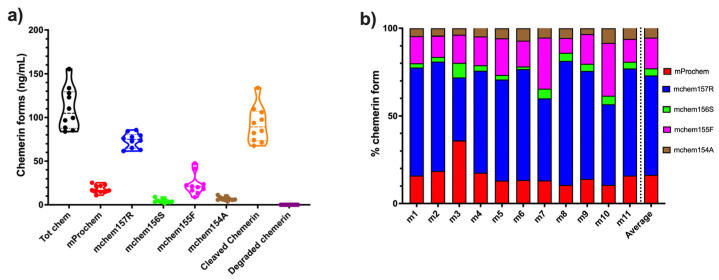
(**a**) Average levels of different forms of mouse chemerin in the ascites harvested from mice (n = 11) with ID8 tumors sampled at sacrifice. Total chemerin and the different chemerin forms were determined by ELISA. Levels of cleaved chemerin were calculated as total chemerin–chem163S, and degraded chemerin calculated as total chemerin–(mProchem + mchem157R + mchem156S + mchem155F + mchem154A). Data shown in a violin plot with thick dotted line representing the median and the thin dotted lines quartiles. (**b**) Levels of different specific chemerin forms in individual mice; shown as % based on data shown in Figure 7a. The average % distribution is shown in the right-most column.

**Figure 9 biomedicines-13-00991-f009:**
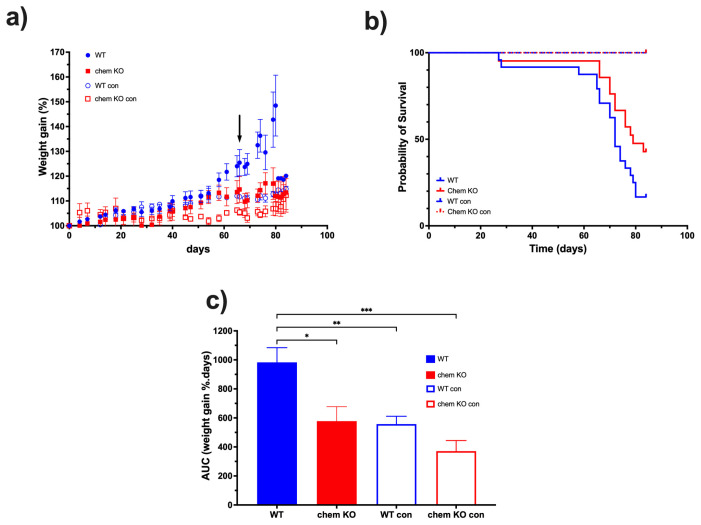
WT and chemerin KO mice were monitored by weight after inoculation with ID8 tumor cells and control mice: WT con and chemerin KO con were injected with PBS. Mice were sacrificed after reaching the allowable maximum weight gain or on Day 84. (**a**) % Weight gain during the survival study. The black arrow represents the endpoint at which mice began to be sacrificed. WT and chemerin KO n = 8 (**b**) Kaplan–Meier curve of the survival of WT (n = 24), chemerin KO (chem KO; n = 21) mice, WT con (n = 8), and chemerin KO con (n = 8) mice. (**c**) Area under the curve (AUC) of weight gain shown as mean ± SEM from the data in (**b**) as described in Materials and Methods, Section 2.6. Data was analyzed by ANOVA followed by post hoc Tukey-Kramer test. * *p* < 0.05, ** *p* < 0.01, *** *p* < 0.001.

**Table 1 biomedicines-13-00991-t001:** Demographics of participants. Volunteers with normal BMI (<25) are shown as <25 BMI and bariatric surgery patients with obesity are shown as >40 BMI.

	<25 BMI	>40 BMI
Number	22	22
Age (mean ± SEM)	47 ± 0.4	47 ± 1.7
Gender (M, F; %M)	16 M, 6 F (73%)	16 M, 6 F (73%)
BMI (mean ± SEM)	23 ± 1.2	46 ± 2.3
Race/ethnicity		
white	6	17
Not Hispanic/Latino	4	2
African American	0	1
American Indian Alaskan native	0	1
Asian	12	0
Declined to answer	0	1

**Table 2 biomedicines-13-00991-t002:** Levels of the different forms of chemerin in plasma and serum from people with <25 BMI (N = 22). The levels of the different chemerin forms were determined by specific ELISA, as described in Section 2.5, and the averages of plasma levels compared to serum levels by two-tailed paired *t* test with coefficient of variance (CV) calculated. N.D.: not detectable. S.D. standard deviation.

	Plasma		Serum		
	Mean ± SD (ng/mL)	CV (%)	Mean ± SD (ng/mL)	CV (%)	*p* Value
Total chemerin	69.9 ± 13.3	19.0	90.9 ± 25.2	27.7	0.0014
chem163S	59.5 ± 11.1	18.6	58.4 ± 10.4	17.8	0.6990
chem158K	3.31 ± 2.46	74.2	2.48 ± 2.65	107.0	0.0174
chem157S	1.97 ± 1.15	58.1	1.98 ± 1.66	84.1	0.9815
chem156F	6.41 ± 3.95	61.6	8.87 ± 4.4	49.6	0.0859
chem155A	1.02 ± 0.947	93.2	1.82 ± 1.14	62.6	<0.0001
Cleaved Chemerin	11.2 ± 11.2	100.0	32.5 ± 23.9	73.6	0.0004
Degraded chemerin	N.D.		19.3 ± 20.5	106.0	0.0009

**Table 3 biomedicines-13-00991-t003:** Levels of the different forms of chemerin in plasma and serum from people with >40 BMI (N = 22). The levels of the different chemerin forms were determined by specific ELISA, as described in Section 2.5, and the averages of plasma levels compared to serum levels by two-tailed paired *t* test with coefficient of variance (CV) calculated. S.D.: standard deviation.

	Plasma		Serum		
	Mean ± SD (ng/mL)	CV (%)	Mean ± SD (ng/mL)	CV (%)	*p* Value
Total chemerin	90.8 ± 18.4	20.3	113 ± 24.4	21.6	<0.0001
chem163S	49.4 ± 20.4	41.3	52.5 ± 14.2	27.1	0.4169
chem158K	3.73 ± 3.09	82.8	1.67 ± 1.99	119	0.0015
chem157S	3.45 ± 1.08	31.2	6.19 ± 1.7	27.4	<0.0001
chem156F	24.4 ± 14.2	58.3	10.8 ± 7.44	68.9	0.0005
chem155A	2.59 ± 1.06	40.7	6.34 ± 2.95	46.5	<0.0001
Cleaved Chemerin	41.4 ± 25.9	62.7	60.5 ± 27.1	44.8	0.0049
Degraded chemerin	10.5 ± 11.6	111	36.8 ± 21.6	58.6	<0.0001

**Table 4 biomedicines-13-00991-t004:** Higher chemerin levels in plasma and serum from people with obesity (BMI > 40; N = 22) than volunteers (BMI < 25; N = 22). The levels of the different chemerin forms were determined by specific ELISA, as described in Section 2.5, and the average fold change in plasma and serum levels (fold change = level in BMI > 40/level in BMI < 25) compared between volunteers (data in Table 2) and people with obesity (Data in Table 3). The data was analyzed by two-tailed paired *t* test. N.A.: not applicable.

	Plasma		Serum	
	Fold Change	*p* Value	Fold Change	*p* Value
Total chemerin	1.30	0.0001	1.24	0.0043
chem163S	0.83	0.0428	0.90	0.1114
chem158K	1.13	0.6140	0.67	0.2473
chem157S	1.75	<0.0001	3.13	<0.0001
chem156F	3.80	<0.0001	1.22	0.2921
chem155A	2.55	<0.0001	3.48	<0.0001
Cleaved Chemerin	3.98	<0.0001	1.86	0.0006
Degraded chemerin	N.A.		2.05	0.0120

**Table 5 biomedicines-13-00991-t005:** Demographics of participants with ovarian carcinoma (OC patients; N = 22).

	OC Patients
Number	22
Age (mean ± SEM)	63 ± 1.6
Prior BMI (mean ± SEM)	28 ± 1.4
BMI at collection	29 ± 1.7
Serous ovarian carcinoma	8
Fallopian tube serous carcinoma	4
Primary peritoneal serous carcinoma	3
Clear cell ovarian carcinoma	1
De-differentiated ovarian carcinoma	1
Mucinous borderline ovarian carcinoma	1
Gastric signet cancer	1
cervical squamous adenocarcinoma	1
Uterine carcinosarcoma	1
benign serous cyst, para-ovarian	1
Race/ethnicity	
White	11
Hispanic/Latina	6
African American	3
Asian	1
Declined to answer	1

**Table 6 biomedicines-13-00991-t006:** Levels of the different forms of chemerin in ascitic fluid from ovarian carcinoma patients. The levels of the different chemerin forms were determined by specific ELISA, as described in Section 2.5, with coefficient of variance (CV) calculated. S.D. standard deviation. N = 16 except for total chemerin where N = 22, chem158K, chem157S and chem156F where N = 19.

	Mean ± SD (ng/mL)	CV (%)
Total chemerin	48.3 ± 26.1	54.1
chem163S	16.2 ± 26.1	162
chem158K	18.7 ± 14.7	81.2
chem157S	19.0 ± 12.3	64.6
chem156F	14.3 ± 21.4	150
chem155A	11.4 ± 6.90	60.4
Cleaved Chemerin	40.5 ± 28.6	70.5
Degraded chemerin	5.67 ± 34.4	12.1

**Table 7 biomedicines-13-00991-t007:** Levels of the different forms of mouse chemerin in ascitic fluid of mice with ID8 tumors. The levels of the different chemerin forms were determined by specific ELISA for mouse chemerin forms, as described in Section 2.5, with coefficient of variance (CV) calculated. N.A.: not applicable. S.D. standard deviation.

	Mean ± SD (ng/mL)	CV (%)
Total chemerin	110 ± 24	21.7
mProchem	18.2 ± 4.62	25.4
mchem157R	73.9 ± 8.43	11.4
mchem156S	4.51 ± 2.05	45.4
mchem155F	23.4 ± 12.1	51.6
mchem154A	6.84 ± 2.07	30.3
Cleaved Chemerin	92.1 ± 20.5	22.3
Degraded chemerin	0 ± 0	N.A.

## Data Availability

Data is available upon application to the corresponding author.

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
