# Peer review of "Active Forms of Chemerin Are Elevated in Human and Mouse Ovarian Carcinoma"

_biomedicines, 2025, doi:10.3390/biomedicines13040991_

Round 1

Reviewer 1 Report

Comments and Suggestions for Authors

Dear authors;

About a fifth of references are self-citation and you embedded it in everywhere of your manuscript, please reduce them and removed unnecessary references. Also, there are some similar articles. It can reduce your novelty.  below I provide my other commentaries.

Title:

  • What is the novelty of your work? There are some articles with similar title. Please check and change it to highlight your novelty.

Abstract:

  • Line 14-16, what about BMI 25-39?
  • Line 18-19, please revise this sentence to the clear one.
  • Did you analyze BMI in ovarian carcinoma ascites cases?
  • Please write the conclusion more clearly. It seems that you just wrote results and finished it with “which might be enhancing tumor growth” as conclusion.

Introduction:

  • As you mentioned in line 52-53, there are some references that indicated the level of chemerin is higher in ascitic fluid than in serum, also I can find other references on other years. why should you do that again, what is your novelty?
  • Line 68-70, what was your aim to find similar chemerin form in mice? Animal study usually design before human researches.
  • In addition, there are some references that show there is a correlation between BMI and chemerin concentration.

Materials and Methods:

Section 2.1

  • Line 77, the range of volunteers was 18-80. As we know age can affect in chemerin concentration. You should mention to these tips, as well as BMI.
  • Did you have ethical code? Please, add it.

Section 2.1 (you have similar sections)

  • I think you could check the BMI in ovarian carcinoma ascites samples.
  • Please correct the numbers of all sections

Section 3.1

  • Do you have any ethical code for animal study?
  • Please add your replication repeat.

Results:

  • Thanks for your valuable table1. Does the race can affect on your result? Also, check the “Table A1” in line 146. I think it is Table 1 not Table A1.
  • I cannot understand your self-citations in the result. There are unnecessary. Please remove them.
  • Line 149-153, some methods com in results section, move it to its appropriate section.

Discussion:

  • Line 400-403, there are no correlation between your conclusion and title.
  • Based on your work, why does serum have the more chemerin than plasma?

Author Response

Reviewer 1

Dear authors;

About a fifth of references are self-citation and you embedded it in everywhere of your manuscript, please reduce them and removed unnecessary references. Also, there are some similar articles. It can reduce your novelty.  below I provide my other commentaries.

The authors thank the reviewer for their comments. The percentage of self-citations is <15%.

Title:

  • What is the novelty of your work? There are some articles with similar title. Please check and change it to highlight your novelty.

 The title has been modified to: “Active forms of chemerin are elevated in human and mouse ovarian carcinoma

Abstract:

  • Line 14-16, what about BMI 25-39?
  • Participants were deliberately recruited to be in the 18-25 BMI (normal BMI) and >40 (morbidly obese BMI) to clarify differences in chemerin levels and processing between these two groups. Future studies will investigate chemerin levels in the overall population.
  • Line 18-19, please revise this sentence to the clear one.
  • Sentence has been edited to read: “Conversion of plasma to serum increased the levels of processed chemerin with lower increases in samples from people with BMI <25 than in people >40 BMI.”
  • Did you analyze BMI in ovarian carcinoma ascites cases?
  • The data is in Table 5 showing that the prior BMI was 28 ± 1.4 and was 29 ± 1.7 at time of collection.
  • Please write the conclusion more clearly. It seems that you just wrote results and finished it with “which might be enhancing tumor growth” as conclusion.
  • The final sentence has been edited to read:” Ascites of ovarian carcinoma patients contained high levels of active chemerin which, based on the mouse data, enhance tumor growth.”

Introduction:

  • As you mentioned in line 52-53, there are some references that indicated the level of chemerin is higher in ascitic fluid than in serum, also I can find other references on other years. why should you do that again, what is your novelty?
  • A new sentence has been added to the end of the next paragraph to describe the previously used ELISAs.
  • Line 68-70, what was your aim to find similar chemerin form in mice? Animal study usually design before human researches.
  • This study was not designed as a translational study but instead was conceived as a bedside to bench investigation.
  • In addition, there are some references that show there is a correlation between BMI and chemerin concentration.
  • This study is the first to systematically investigate differences between plasma and serum levels of chemerin. Because of the known differences in chemerin levels with BMI, participants were enrolled as <25 BMI and >40 BMI to determine if changes in chemerin levels and forms were affected by the conversion of plasma to serum was affected by BMI.

Materials and Methods:

Section 2.1

  • Line 77, the range of volunteers was 18-80. As we know age can affect in chemerin concentration. You should mention to these tips, as well as BMI.
  • A new paragraph (lines 428 -438) has been added to discuss in more detail demographic confounding factors.
  • Did you have ethical code? Please, add it.
  • Sentence on lines 77 and 87 onwards describes IRB approvals for these studies.

Section 2.1 (you have similar sections)

  • I think you could check the BMI in ovarian carcinoma ascites samples.
  • The data is in Table 5 showing that the prior BMI was 28 ± 1.4 and was 29 ± 1.7 at time of collection.
  • Please correct the numbers of all sections
  • We were unable to find two Sections 2.1.

Section 3.1

  • Do you have any ethical code for animal study?
  • Sentence on line 186 onwards describes IRB approvals for these studies.
  • Please add your replication repeat.
  • Sentence on line 203 onwards describes that the mouse experiments were performed three times.

Results:

  • Thanks for your valuable table1. Does the race can affect on your result? Also, check the “Table A1” in line 146. I think it is Table 1 not Table A1.
  • Table 1 is the table in the main text and covers the demography of the participants. Table A1 is supplementary material containing details of medications. Race as a confounding factor is discussed in the new paragraph (lines 547 -559).
  • I cannot understand your self-citations in the result. There are unnecessary. Please remove them.
  • Line 149-153, some methods com in results section, move it to its appropriate section.
  • We included these brief notes on how the experiments were conducted to make it easier for the reader to follow.

Discussion:

  • Line 400-403, there are no correlation between your conclusion and title.
  • Based on your work, why does serum have the more chemerin than plasma?
  • When serum is generated from plasma proteases are activated which cleave different chemerin forms changing the distribution between the forms. The apparent levels of total chemerin increases when plasma is converted to serum because the ELISA used to determine total chemerin has different sensitivity for different forms (Chang et al 2016). In addition, there may be a contribution from platelets discussed on lines 526 – 530.

Reviewer 2 Report

Comments and Suggestions for Authors

Principal Issues:
Biological Importance of Chemerin Stimulation in Ovarian Cancer:
The research indicates a link between elevated active chemerin levels and tumor proliferation, although the molecular understanding is insufficient. Elaborating on the molecular processes, including the roles of the PI3K/AKT and MAPK signaling pathways and their subsequent impacts on tumor growth, migration, or immunological regulation, would enhance the study's significance. Further exploration of the putative signaling pathways implicated in tumor growth would augment the significance.
Incorporating immunohistochemistry or functional tests would be advantageous to validate the tumor-promoting effect of chemerin in ovarian cancer cells.
Comparative Analysis of Human and Murine Models:
The differences in chemerin isoforms and their functionality between humans and mice must be carefully addressed, especially whether the same proteolytic enzymes mediate chemerin activation in both species. This would enhance the translational significance of the results. Are the proteolytic enzymes involved in chemerin stimulation the same in both species?
The text might be improved by a more extensive discussion of the mouse model's translational significance to human illness.
The Function of Chemerin in the Tumor Microenvironment:
The research reveals elevated concentrations of active chemerin in ascitic fluid; nevertheless, it is uncertain whether chemerin influences tumor cells directly or alters the immunological microenvironment. The study should investigate the role of chemerin in immune cell recruitment, specifically regarding macrophage or neutrophil infiltration, and its potential impact on tumor growth.
The paper must examine whether chemerin-induced immune responses, such as macrophage or neutrophil infiltration, facilitate tumor growth.
Statistical Robustness and Sample Size
The sample size for patients with human ovarian cancer is quite small (N=22), thus limiting the research's statistical power. To guarantee the results' robustness, the authors must evaluate the feasibility of further validation in a larger cohort and elucidate the management of any outliers in the statistical analysis. The research may lack sufficient power to draw definitive conclusions, notwithstanding the observed patterns.
Were there any outliers, and what methods were used to address them in the statistical analysis?
Clinical Implications and Prospective Trajectories:
The paper must include possible therapeutic implications, including the targeting of chemerin signaling as an anti-cancer approach. An assessment of current or prospective chemerin-targeting therapeutics, including small molecule inhibitors, monoclonal antibodies, or receptor antagonists, would enhance the clinical significance of the research. Furthermore, evaluating the viability of these methodologies in ovarian cancer therapy, grounded on existing preclinical or clinical data, would provide significant insights for forthcoming research.
Future studies should focus on possible inhibitors of chemerin signaling or its proteolytic activation.
Marginal Issues:
Lucidity and Comprehensibility:
Certain statements are complex and may be rephrased for enhanced clarity.
Typographical mistakes, such as "chem163" in lieu of "chem163S" in some sections, need to be corrected.
The introduction must succinctly encapsulate previous studies on chemerin in various tumors to provide a comprehensive background.
Figure Legends and Tables:
The tales need to be more detailed, offering context for lay readers.
Verify the table formatting for uniformity in space and layout.
Ethical Considerations:
The paper indicates that ethical permissions were secured; nonetheless, it would be beneficial to specify the number of patient samples removed, if any, owing to quality issues.
The work significantly enhances the comprehension of chemerin's function in ovarian cancer. Addressing molecular findings, improving the clarity of comparisons between human and mouse models, and detailing interactions within the tumor microenvironment would substantially fortify the text. These adjustments may enhance the study's capacity to elucidate chemerin's function in cancer biology and its therapeutic significance.

Comments on the Quality of English Language

The English could be improved to more clearly express the research.

Author Response

Reviewer 2

The authors thank the reviewer for their comments and have replied to the individual points below.

principal Issues:
Biological Importance of Chemerin Stimulation in Ovarian Cancer:
The research indicates a link between elevated active chemerin levels and tumor proliferation, although the molecular understanding is insufficient. Elaborating on the molecular processes, including the roles of the PI3K/AKT and MAPK signaling pathways and their subsequent impacts on tumor growth, migration, or immunological regulation, would enhance the study's significance. Further exploration of the putative signaling pathways implicated in tumor growth would augment the significance.

The authors thank the reviewer for their comments and agree that future studies should investigate the mechanism of chemerin in enhancing tumor growth. The paragraph starting at line 693 has been extended with more comments to include these points.

Incorporating immunohistochemistry or functional tests would be advantageous to validate the tumor-promoting effect of chemerin in ovarian cancer cells.

The authors agree that the next steps would be investigate chemerin effects in enhancing ovarian carcinoma growth.

Comparative Analysis of Human and Murine Models:
The differences in chemerin isoforms and their functionality between humans and mice must be carefully addressed, especially whether the same proteolytic enzymes mediate chemerin activation in both species. This would enhance the translational significance of the results. Are the proteolytic enzymes involved in chemerin stimulation the same in both species?
From our studies the proteolytic processing of mouse and human chemerin is very similar. Mouse chemerin is one amino acid shorter than human chemerin and the sequence in the C-terminal tail which interacts with the receptors is highly homologous between species.

The text might be improved by a more extensive discussion of the mouse model's translational significance to human illness.

The discussion has been edited with some comments about translation of the Id8 mouse model to human disease (lines 622 - 628)
The Function of Chemerin in the Tumor Microenvironment:
The research reveals elevated concentrations of active chemerin in ascitic fluid; nevertheless, it is uncertain whether chemerin influences tumor cells directly or alters the immunological microenvironment. The study should investigate the role of chemerin in immune cell recruitment, specifically regarding macrophage or neutrophil infiltration, and its potential impact on tumor growth. The paper must examine whether chemerin-induced immune responses, such as macrophage or neutrophil infiltration, facilitate tumor growth.

The authors thank the reviewers for these comments and agree that these are important questions for future studies. The discussion has been edited with comments about translation of the Id8 mouse model to human disease (lines 623 - 628) and a sentence has been added to the last paragraph of the discussion on future studies (lines 703 - 708).
Statistical Robustness and Sample Size
The sample size for patients with human ovarian cancer is quite small (N=22), thus limiting the research's statistical power. To guarantee the results' robustness, the authors must evaluate the feasibility of further validation in a larger cohort and elucidate the management of any outliers in the statistical analysis. The research may lack sufficient power to draw definitive conclusions, notwithstanding the observed patterns.
Were there any outliers, and what methods were used to address them in the statistical analysis?
Prior to the analysis we had defined an outlier as having a value > 3SDs away from the mean. No points met that criterion so there were no outliers and all available data points were included in the analysis. This has been added to the statistics section (Line 219)

Clinical Implications and Prospective Trajectories:
The paper must include possible therapeutic implications, including the targeting of chemerin signaling as an anti-cancer approach. An assessment of current or prospective chemerin-targeting therapeutics, including small molecule inhibitors, monoclonal antibodies, or receptor antagonists, would enhance the clinical significance of the research. Furthermore, evaluating the viability of these methodologies in ovarian cancer therapy, grounded on existing preclinical or clinical data, would provide significant insights for forthcoming research.

A paragraph has been added to the discussion (lines 683 -693) to review the current status of chemerin antagonist development.
Future studies should focus on possible inhibitors of chemerin signaling or its proteolytic activation.

See responses above.
Marginal Issues:
Lucidity and Comprehensibility:
Certain statements are complex and may be rephrased for enhanced clarity.
Typographical mistakes, such as "chem163" in lieu of "chem163S" in some sections, need to be corrected.

Corrected
The introduction must succinctly encapsulate previous studies on chemerin in various tumors to provide a comprehensive background.

Figure Legends and Tables:
The tales need to be more detailed, offering context for lay readers.

The Table legends have been edited to make them easier to understand.
Verify the table formatting for uniformity in space and layout.

The tables were created in the BioMedicines template so their format is imposing the apparent difficulties with space and layout.
Ethical Considerations:
The paper indicates that ethical permissions were secured; nonetheless, it would be beneficial to specify the number of patient samples removed, if any, owing to quality issues.

  • Sentence on lines 77 and 87 onwards describes IRB approvals for these studies.

The work significantly enhances the comprehension of chemerin's function in ovarian cancer. Addressing molecular findings, improving the clarity of comparisons between human and mouse models, and detailing interactions within the tumor microenvironment would substantially fortify the text. These adjustments may enhance the study's capacity to elucidate chemerin's function in cancer biology and its therapeutic significance.

We have added additional information about other studies as well as a more specific discussion about potential roles of chemerin in the tumor micro-environment to the discussion.

Comments on the Quality of English Language

The English could be improved to more clearly express the research.

Reviewer 3 Report

Comments and Suggestions for Authors

L8: Division of Gynecologic Oncology, Department of Obstetrics and Gynecology, Stanford Women’s Cancer 8 Center, Stanford Cancer Institute, Stanford, CA, USA. Please make font consistent

L38: Ovarian cancer most commonly occurs in women after 38 menopause.  Reference is needed her

L59: Biopsy samples contain mixtures of these forms. Please review location of this sentence!

L64: As controls, plasma and serum from volunteers were 64 assayed for levels of the different chemerin forms. Factoring in the direct-correlation of 65 obesity to the high levels of chemerin available in circulation [18, 19], we divided the 66 donors into two cohorts based on a BMI less than 25 versus bariatric surgery patients with 67 BMI greater than 40. Furthermore, we also investigated if there was similar activation of 68 chemerin in a mouse model of ovarian carcinoma by measuring the levels of different 69 chemerin forms and the co-relation with disease outcome. Consider moving this paragraph to methods

L73: Blood was collected from volunteers with BMI <25 and from bariatric surgery patients 73 (people with obesity; BMI >40) into either serum or plasma tubes. All studies were 74 approved by the Stanford Institutional Review Board under protocol numbers #6946 and 75 #24175 and all individuals were consented to participate in these studies. Please review the structure of the paragraph. Consider starting with All studies were 74 approved…

L84: (Roche Applied Science, Pleasanton, CA, USA). This mixture was shaken at 4°C for. Please check font

L89: The collection of ascitic fluid from ovarian carcinoma patients was approved by the 89 Stanford Institutional Review Board under protocol numbers #42966. Please consider putting approvals together. More information is needed on the collection of ovarian carcinoma ascites samples

L100: All 100 procedures were approved by the institutional animal care and committee of VAPAHCS. All approvals may go into one paragraph 

L171: What is the significance of calculating coefficient of variance (CV) in table 2 and 3?

L265: They are also different from the levels in synovial fluid and cerebrospinal fluid where chem158K, the inactive partially processed form, is the major form and in plasma 266 where the intact precursor predominates. This statement came out of the blue in the result section! Please give more information in the introduction and consider to move to discussion!

L 365: The levels of the different chemerin forms were determined by ELISA with coefficient of variance (CV) calculated. What is the significance of calculating CV?

L 393: Chemerin deficient mice have reduced ovarian carcinoma in this model suggesting that chemerin is enhancing tumor growth. This is an important statement that need further discussion which the authors mentioned in discussion. Please consider move the statement to discussion

L474: regulates lipid metabolism. What is the relevant of lipid metabolism regulation to Chemerin? Probably related to clear cell renal carcinoma progression as per Ref 44?

L22 and 514: Overall this study shows that chemerin enhances ovarian tumor growth in both human and mouse. In L480: Chemerin can also inhibit tumor growth as demonstrated in studies including melanoma and hepatocellular carcinoma. In the B16 cell mouse model of melanoma, chemerin expression reduced tumor growth in vivo but not in vitro showing that the effect required the host immune system. The authors may comment in the reason why Chemerin enhances ovarian tumor growth and inhibit the growth of other tumors as discussed in ref 54.

Author Response

Reviewer 3

The authors thank the reviewer for their comments.

L8: Division of Gynecologic Oncology, Department of Obstetrics and Gynecology, Stanford Women’s Cancer 8 Center, Stanford Cancer Institute, Stanford, CA, USA. Please make font consistent

Corrected

L38: Ovarian cancer most commonly occurs in women after 38 menopause.  Reference is needed her

Reference #5 has been added.

L59: Biopsy samples contain mixtures of these forms. Please review location of this sentence!

Sentence has been moved and merged with the sentence that now starts on line 67

L64: As controls, plasma and serum from volunteers were 64 assayed for levels of the different chemerin forms. Factoring in the direct-correlation of 65 obesity to the high levels of chemerin available in circulation [18, 19], we divided the 66 donors into two cohorts based on a BMI less than 25 versus bariatric surgery patients with 67 BMI greater than 40. Furthermore, we also investigated if there was similar activation of 68 chemerin in a mouse model of ovarian carcinoma by measuring the levels of different 69 chemerin forms and the co-relation with disease outcome. Consider moving this paragraph to methods

That paragraph has been split between the introduction and Materials and Methods line 87 - 92

L73: Blood was collected from volunteers with BMI <25 and from bariatric surgery patients 73 (people with obesity; BMI >40) into either serum or plasma tubes. All studies were 74 approved by the Stanford Institutional Review Board under protocol numbers #6946 and 75 #24175 and all individuals were consented to participate in these studies. Please review the structure of the paragraph. Consider starting with All studies were 74 approved…

Corrected.

L84: (Roche Applied Science, Pleasanton, CA, USA). This mixture was shaken at 4°C for. Please check font

Font is all Palatino linotype 10

L89: The collection of ascitic fluid from ovarian carcinoma patients was approved by the 89 Stanford Institutional Review Board under protocol numbers #42966. Please consider putting approvals together. More information is needed on the collection of ovarian carcinoma ascites samples

The section has been modified to read:

2.1 Human ovarian carcinoma ascites samples

The collection of ascitic fluid from ovarian carcinoma patients was approved by the Stanford Institutional Review Board under protocol numbers #42966. Ascites was directly collected from patients at the time of surgery or by paracentesis and transferred to lab for post processing. After centrifugation of ascites samples at 300g for 10 minutes, cell free supernatant was immediately separated, aliquoted, and stored at -80°C for further use. Upon thawing, protease inhibitors (Complete Protease Inhibitor, Roche Applied Science, Pleasanton, CA, USA) were added to the vials, while in a few samples, protease inhibitors were added immediately upon collection of ascitic fluid. The samples underwent one freeze-thaw cycle prior to chemerin measurements.

L100: All 100 procedures were approved by the institutional animal care and committee of VAPAHCS. All approvals may go into one paragraph 

The approvals for both human and animal work have been edited – see responses to reviewers 1 and 2.

L171: What is the significance of calculating coefficient of variance (CV) in table 2 and 3?

The coefficient of variance was calculated to investigate if there was less variability in plasma or serum and between people with <25 BMI compared to people with >40 BMI. No consistent differences were found across all of the assays but some chemerin forms such as chem158K have much larger variances than others such as the total chemerin ELISA.

L265: They are also different from the levels in synovial fluid and cerebrospinal fluid where chem158K, the inactive partially processed form, is the major form and in plasma 266 where the intact precursor predominates. This statement came out of the blue in the result section! Please give more information in the introduction and consider to move to discussion!

That sentence has been moved to the discussion (paragraph starting line 532)

L 365: The levels of the different chemerin forms were determined by ELISA with coefficient of variance (CV) calculated. What is the significance of calculating CV?

The CV was calculated for the different chemerin forms to provide a comparator for series of samples where the biological variance due to genetics, medical history type of cancer etc was minimized. This has now been expanded in the discussion (lines 564 -567).

L 393: Chemerin deficient mice have reduced ovarian carcinoma in this model suggesting that chemerin is enhancing tumor growth. This is an important statement that need further discussion which the authors mentioned in discussion. Please consider move the statement to discussion

The sentence summarizes the conclusion from the mouse experiments and is also discussed further in lines 623 onwards which expand on that conclusion.

L474: regulates lipid metabolism. What is the relevant of lipid metabolism regulation to Chemerin? Probably related to clear cell renal carcinoma progression as per Ref 44?

A hypothesis is that chemerin regulates energy balance and alters lipid metabolism which may be linked to the Warburg effect of energy utilization in cancers. This might be a second mechanism by which chemerin modulates tumor growth in addition to its effects on the immune system.

L22 and 514: Overall this study shows that chemerin enhances ovarian tumor growth in both human and mouse. In L480: Chemerin can also inhibit tumor growth as demonstrated in studies including melanoma and hepatocellular carcinoma. In the B16 cell mouse model of melanoma, chemerin expression reduced tumor growth in vivo but not in vitro showing that the effect required the host immune system. The authors may comment in the reason why Chemerin enhances ovarian tumor growth and inhibit the growth of other tumors as discussed in ref 54.

More discussion has been added on chemerin’s role in regulating the tumor micro-environment.

Round 2

Reviewer 1 Report

Comments and Suggestions for Authors

Reviewer 1

Dear authors;

Thank you for your appropriate editing and clear answers. Below I provide my other commentaries.

The authors thank the reviewer for their comments. The percentage of self-citations is <15%.

  • Thank you. I will leave the decision on this to the editorial team.

Materials and Methods:

Section 2.1

  • Line 77, the range of volunteers was 18-80. As we know age can affect in chemerin concentration. You should mention to these tips, as well as BMI.

A new paragraph (lines 428 -438) has been added to discuss in more detail demographic confounding factors.

  • Thank you for your new paragraph, it is in line 548-560 in new edit. Based on your new information, you have worked on the age group of 18 to 80 years old. It is better to discuss about age effect on your results.

Section 2.1 (you have similar sections)

  • Please correct the numbers of all sections

We were unable to find two Sections 2.1.

  • But as I see in new version you have edited it.

  • I cannot understand your self-citations in the result. There are unnecessary. Please remove them.
  • Are these self-citations necessary?

Author Response

Reviewer 1

Dear authors;

Thank you for your appropriate editing and clear answers. Below I provide my other commentaries.

The authors thank the reviewer for their detailed reading of the manuscript.

The authors thank the reviewer for their comments. The percentage of self-citations is <15%.

  • Thank you. I will leave the decision on this to the editorial team.

 Please see response to last comment below.

Materials and Methods:

Section 2.1

  • Line 77, the range of volunteers was 18-80. As we know age can affect in chemerin concentration. You should mention to these tips, as well as BMI.

A new paragraph (lines 428 -438) has been added to discuss in more detail demographic confounding factors.

  • Thank you for your new paragraph, it is in line 548-560 in new edit. Based on your new information, you have worked on the age group of 18 to 80 years old. It is better to discuss about age effect on your results.

 Thanks for this comment. We think that the issue of age fits better in the discussion than results as age what explicitly studied here. To our knowledge there have been no longitudinal studies on the levels of chemerin or its various forms so it is not possible to make any more specific points about age as a potential confounding factor.

Section 2.1 (you have similar sections)The citations

  • Please correct the numbers of all sections

We were unable to find two Sections 2.1.

  • But as I see in new version you have edited it.

The section numbering is consistent. 

  • I cannot understand your self-citations in the result. There are unnecessary. Please remove them.
  • Are these self-citations necessary?

The citations are needed for a reader to understand the background of this study and self-citation is necessary because our lab has been pioneering the study of chemerin forms in both human and mouse and has been responsible for the majority of papers in this field.

Reviewer 2 Report

Comments and Suggestions for Authors

I thank the authors for submitting their responses to the reviewers, their revised manuscript, and their research efforts. 

Author Response

I thank the authors for submitting their responses to the reviewers, their revised manuscript, and their research efforts. 

The authors thank the reviewer for their kind comments.